# Homologous Recombination: To Fork and Beyond

**DOI:** 10.3390/genes9120603

**Published:** 2018-12-04

**Authors:** Félix Prado

**Affiliations:** Department of Genome Biology, Andalusian Molecular Biology and Regenerative Medicine Center (CABIMER), CSIC-University of Seville-University Pablo de Olavide, 41092 Seville, Spain; felix.prado@cabimer.es

**Keywords:** homologous recombination, replication stress, DNA damage tolerance, fork stability, single-strand DNA gap filling

## Abstract

Accurate completion of genome duplication is threatened by multiple factors that hamper the advance and stability of the replication forks. Cells need to tolerate many of these blocking lesions to timely complete DNA replication, postponing their repair for later. This process of lesion bypass during DNA damage tolerance can lead to the accumulation of single-strand DNA (ssDNA) fragments behind the fork, which have to be filled in before chromosome segregation. Homologous recombination plays essential roles both at and behind the fork, through fork protection/lesion bypass and post-replicative ssDNA filling processes, respectively. I review here our current knowledge about the recombination mechanisms that operate at and behind the fork in eukaryotes, and how these mechanisms are controlled to prevent unscheduled and toxic recombination intermediates. A unifying model to integrate these mechanisms in a dynamic, replication fork-associated process is proposed from yeast results.

## 1. Introduction

The homologous recombination (HR) machinery uses intact DNA molecules as a template to repair DNA breaks. For many years, research about HR focused on the repair of double-strand breaks (DSBs) in both mitosis and meiosis; accordingly, most of our knowledge about the proteins, activities, and mechanisms of HR comes from these specific repair events [1,2]. Indeed, the fact that both spontaneous and induced (either by genotoxic agents or mutations) recombination depend on the same genetic factors as DSB-induced recombination led to the long-held idea that DSBs are a common intermediate in most of those events. Three stages are critical during recombinational repair of DSBs for providing the means to search for and then invade an intact homologous template: (i) DNA resection of the 5′-ends of the DSB, which generates 3′-ended, single-strand DNA (ssDNA) molecules; (ii) ssDNA annealing; and (iii) DNA strand exchange; invasion of the template leads to the formation of a D-loop that is further stabilized by DNA synthesis (Figure 1A). In eukaryotes, resection is carried out by a battery of proteins, including the nucleases Mre11 and Exo1, the helicase Sgs1 (yeast)/RECQ (human) and the helicase/nuclease Dna2 [3]. The classical HR proteins Rad52, Rad51, Dmc1 (meiosis-specific Rad51 paralog), and the helicase Rad5 (yeast)/HTLF (human) can perform ssDNA annealing and DNA strand exchange, whereas yeast Rad59 can promote ssDNA annealing [4,5,6,7,8,9]. In yeast, Rad52 plays an additional mediator role by facilitating the formation of the Rad51/ssDNA nucleofilament required to search for and invade the homologous DNA sequence; this function is carried out by BRCA2 in humans [10]. The D-loop intermediate is processed through different pathways that can lead to crossovers between the broken and template molecules (Figure 1A). When the recombinogenic process leads to the formation of a double-Holliday junction (HJ), this intermediate is either dissolved by the STR complex (formed by Sgs1, the topoisomerase III (Top3), and the accessory factor Rmi1 in yeast and their orthologs BLM, TOPOIIIα, and RMI1 and RMI2 in humans) [11] or resolved by specific nucleases [12].

Over the last few years, it has become clear that the recombination proteins have additional and essential functions in response to replicative stress that are not associated with DSB repair. Specifically, a number of these functions are required both to bypass DNA lesions that hamper the advance of the replicative polymerase and to fill in the fragments of ssDNA generated during this process (Figure 1B). Although replication fork bypass of blocking lesions facilitates the timely completion of DNA replication by postponing their repair, it also challenges genome integrity, as it can lead to mutagenesis and genomic rearrangements. Thus, the proteins involved in these DNA damage tolerance (DDT) mechanisms are tightly regulated to prevent unscheduled and toxic DNA intermediates. In particular, HR during DDT provides an alternative, error-free mechanism to translesion synthesis (TLS), during which specialized polymerases incorporate a nucleotide opposite to the lesion that can result in high mutagenesis rates (see [13] for a recent review on TLS). In contrast, HR uses the intact sister chromatid to circumvent the lesion, and it is believed to occur using two pathways that are usually referred to as the template switching and salvage pathways (Figure 1B) [14,15].

We are still far from understanding how HR and other DDT proteins deal with replicative DNA lesions. Here, I present a mechanistic overview of the recombination process during DDT in eukaryotes. I dissect the process’ known roles at both the fork and the ssDNA gaps left behind the fork as a consequence of the lesion bypass, as well as the genetic and physical interactions that connect both processes. Finally, I present a unifying model from yeast results, which simplifies the complexity of the multiple pathways and interactions that occur in both scenarios. Nonetheless, many questions remain that now need to be addressed to obtain a better understanding of the genetic consequences of failing in the recombinational response to replicative DNA damage, in particular in human cells.

## 2. Two Different Scenarios for Recombination during DDT: ssDNA Gaps at and behind the Fork

Early works showed that HR proteins are required to fill in the ssDNA lesions generated during DNA replication in the presence of DNA adducts, such as those induced by UV light or the alkylating agent methyl-methane sulfonate (MMS) [16,17,18,19]. The positions of these ssDNA fragments could not be determined in these studies, as they were detected by separating pulse-labelled DNA over alkaline glucose gradients. Yet the finding that HR proteins facilitate the advance of stressed replication forks (Section 3) raised the possibility that lesion-bypass and ssDNA filling-in are mechanistically coupled to the fork. However, single molecule analyses by electron microscopy of DNA molecules from yeast cells and *Xenopus* egg extracts lacking the HR proteins Rad51 or Rad52 and treated with UV light or MMS revealed an accumulation of ssDNA gaps behind the fork [20,21]. This indicates that, at least in these organisms, HR also operates at ssDNA lesions behind the fork and suggests that replicative and repair activities are spatially separated. Indeed, analyses of DNA damage-induced Rad51 containing foci in yeast and mammalian cells showed the existence of both replicative and non-replicative/DNA repair centers [22,23], with the latter restricted to G2/M in yeast through a Mrc1-dependent mechanism that prevents their assembly during S phase [23,24,25,26]. Although the DNA damage and replicative checkpoints sense different types of DNA lesions, they share the molecular signal that triggers the response (an accumulation of ssDNA) as well as essential factors, including the sensor kinases (Mec1/Ddc2 in yeast, and ATR/ATRIP in humans) and the effector kinases (Rad53 and Chk1 in yeast, and CHK1 and CHK2 in humans). One major difference is the mediator protein that amplifies the signal at the DNA damage checkpoint (Rad9 in yeast, 53BP1 in humans) and the replication checkpoint (Mrc1 in yeast, CLASPIN in humans) [27,28]. As the checkpoint mediator Mrc1 preferentially signals replication fork-associated ssDNA [29,30], I have previously suggested that this strategy might ensure that DNA repair centers are not assembled as long as there are stressed forks, as these recombination centers might interfere with proper DNA replication and favor fork-driven genomic rearrangements [31]. Interestingly, the ssDNA fragments left behind the fork upon MMS-induced stress trigger a checkpoint response that is genetically different from the one from stalled fork-associated ssDNA: ssDNA gaps at the fork are signaled by Mrc1, while ssDNA behind the fork are signaled by Rad9 [32,33]. Thus, although both ssDNA lesions, at or behind the fork, stem from the same replicative problem (the encounter of the fork with a blocking lesion), they are signaled through different mechanisms.

## 3. Recombination at the Fork

In response to replication stress, eukaryotic cells accumulate ssDNA gaps at the replication fork, this is believed to be due to the uncoupling between the helicase and polymerase activities of the replisome at the leading strand [20,21,34]. HR proteins assist replication forks through different mechanisms that differ depending on the organisms and stress conditions. Yeast Rad51 and Rad52 can be detected at both unperturbed and MMS-stressed forks, suggesting that the recombinases are not recruited specifically to stalled forks but rather travel with the fork to assist it in response to replication problems [23]. This escort function seems to be conserved in human cells, as different HR proteins (including Rad51) are detected at the nascent chromatin together with the replication machinery [34,35,36,37]. However, human Rad51 slows down DNA synthesis [34,38], whereas yeast Rad51 accelerates it [23,25,39], suggesting different modes of action at blocked forks. Moreover, Rad51 is essential in mammals but not in yeast [40,41], although this difference might reflect a more demanding structural complexity, and accordingly more naturally-occurring replication obstacles, of the higher eukaryote genome.

A reduction in the amount of Rad51 in human cells increases the length of the ssDNA gaps generated by a broad range of genotoxic agents, including MMS and UV light, indicating that Rad51 prevents an excessive accumulation of ssDNA at stressed replication forks [34]. Importantly, human cells respond to these replicative lesions with the formation of reversed forks through a mechanism that, at least for the genotoxic agents camptothecin (CPT), mitomycine (MMC), or hydroxyurea (HU), is completely dependent on Rad51 [34]. Reversed forks were initially proposed to explain replication fork bypass of blocking lesions in mammalian cells; they would be formed upon displacement and reannealing of the nascent strands, leading to a Holliday junction (HJ)–like structure; this structure might facilitate replication fork bypass, by either strand invasion ahead of the fork or DNA synthesis and fork regression, thus reducing the need for post-replicative ssDNA repair (Figure 2) [42]. In contrast to its homolog in bacteria (RecA), Rad51 does not have fork-remodeling activity [43]. Thus, it may control either the recruitment or the activity of fork reversal enzymes (Rad54 and Rad5 in yeast, and HLTF, SMARCAL1, ZRANB3, FANCM, and RECQ in humans) [44,45].

In mammalian cells, BRCA2-dependent loading of Rad51 is required to protect newly synthesized DNA upon replicative stress [46,47,48,49,50,51,52]. The mechanism of protection seems to vary depending on the nature of the lesion. At stalled forks induced by HU, UV, light or CPT, Rad51 protects the newly synthesized DNA from unscheduled and extensive DNA degradation by the nucleases Mre11 and Exo1 [46,47,48,50,51]. In contrast, at stalled forks induced by MMC, Rad51 controls the nuclease activity of Dna2 [49,51]. This difference likely reflects the difficulty in bypassing the DNA interstrand crosslinks generated by MMC, which actually requires a complex network of DNA repair factors, termed the Fanconi anemia (FA) pathway, which includes FA-specific factors, DNA structure-dependent nucleases, and components of the TLS and HR machineries [53].

The Rad51 protective role at stalled forks might be associated with its ability to bind ssDNA and form nucleofilaments. According to this possibility, the stability of the Rad51 nucleofilament is dispensable for DSB repair but essential for replication fork stability [46]. Replication fork stability requires the replication protein A (RPA) complex to coat the ssDNA gaps, and the replicative checkpoint ensures this mechanism by inhibiting replication origins, thus preventing an excess of ssDNA and RPA exhaustion [54]. Since the mediators BRCA2 and Rad52 help Rad51 to compete with RPA at ssDNA [4], it is conceptually possible that Rad51 replaces the protective function of RPA at the fork. Recent studies combining defects in fork reversal enzymes with conditions that reduce or increase the amount of Rad51 at the fork have elegantly shown that the protection of HU-stalled forks against extensive degradation in mammalian cells occurs in two steps: (i) fork reversion through a mechanism that requires Rad51, RPA, and fork reversal enzymes (but BRCA2-independent), and (ii) reversed fork protection against nucleases by additional Rad51 binding (presumably through a stable nucleofilament) (Figure 2). Importantly, both a lack of and an excess of fork reversion lead to genetic instability (see [44,45] for recent and detailed reviews about fork reversion and protection).

Reversed forks can be detected in yeast cells and *Xenopus* egg extracts treated with CPT at similar levels as in human cells [55]. However, they are rare structures in yeast cells treated with MMS or UV light [21,56], except in checkpoint or primase/Ctf4 mutants defective for fork stability or repriming, respectively [57,58,59]. In contrast to MMS or UV light, CPT generates hard-to-bypass lesions by trapping topoisomerase I, as it is covalently linked to a nicked DNA intermediate (Top1 cleavage complex, Top1cc) [60]. Thus, the scarcity of reversed forks in MMS-treated yeast cells might reflect transient structures. Indeed, the helicase Mph1, which promotes fork reversal in vitro, protects MMS-stressed forks in yeast [61]. Along the same line, yeast Rad51 might deal with stalled forks through additional and transient recombinogenic structures. DNA adducts generated by the alkylating agent adozelesin lead to blocked replication forks that are associated with Rad51-dependent HJ structures, which are required for DNA synthesis after the block [62]. However, it is unclear whether these structures are specific for these particular hard-to-bypass lesions or if they are a common intermediate that is resolved once the blocking lesion is circumvented, with this event occurring faster with MMS- or UV-induced lesions.

The absence of Rad51 in yeast cells and *Xenopus* egg extracts leads to the formation of large ssDNA gaps at the replication fork; interestingly, the length and frequency of these ssDNA gaps is not affected by treatment with genotoxic agents like UV light or MMS, which strongly increase the number of replication obstacles [20,21]. This suggests that Rad51 is required to prevent an excess of ssDNA at the fork through a process that does not require activation of the DDT response. Although we still lack information about this function, it is tempting to speculate about the possibility that Rad51 prevents ssDNA from accumulating at the fork by coupling DNA unwinding and DNA synthesis, with this function being even more critical in the presence of DNA damage. In this regard, Rad51 is known to physically interact with the replicative helicase minichromosome maintenance (MCM) in both humans and yeast, and even though its physiological relevance is still unknown [63,64], we recently observed that this interaction is regulated at chromatin by the presence of replicative DNA damage.

## 4. Controlling Recombination at the Fork

Regardless of the mechanism by which Rad51 promotes replication under stress conditions, its activity has to be tightly regulated as excess causes genetic instability [44,65]. In mammalian cells, this control is specifically exerted at the fork by the ssDNA binding protein RADX, which competes with Rad51 to prevent aberrant fork remodeling and genetic instability [66,67]. In addition, cells, from yeast to humans, prevent unscheduled recombination events under unperturbed conditions by the activity of a number of helicases that can disrupt the Rad51 nucleofilament. These helicases include Srs2, Sgs1, and Fbh1 in yeast and their orthologs PARI (Srs2), the RecQ-like helicases BLM, and RECQL5 (Sgs1) and FBH1 in humans [68,69,70,71,72,73]. These activities are tightly regulated to allow Rad51 to operate in response to replication stress. For instance, human cells facilitate HR at MMC-stalled forks through a factor, BODL1, that counteracts the anti-recombinogenic activities of BLM and FBH1, thus stabilizing Rad51 and preventing DNA2-mediated fork over-resection [51].

An additional and conserved mechanism of Rad51 control is mediated by Srs2/PARI. Cells, from yeast to humans, prevent unscheduled recombination events under unperturbed conditions through sumoylation of the replication processivity factor proliferating cell nuclear antigen (PCNA) at lysine 164, which promotes the recruitment of Srs2 in yeast [74,75,76] and PARI in human cells [73] (Figure 3a). This raises the question of how cells relieve Rad51 inhibition at stressed forks to facilitate their replicative activities in response to DNA damage. This mechanism has been recently elucidated in yeast cells treated with MMS. Specifically, it requires the activity of the SUMO-like domain-containing protein Esc2, which binds to stalled forks to locally reduce the levels of Srs2 through two interconnected mechanisms: (i) physical recruitment of the PCNA unloader Elg1, which removes PCNA-bound Srs2; and (ii) physical targeting of Srs2 to the Slx5/Slx8 complex, which promotes Srs2 degradation by the proteasome (Figure 3b) [77]. It has been shown biochemically that Srs2 and PARI can prevent HR not only through disruption of Rad51 nucleofilaments [71,72,73] but also through inhibition of DNA repair synthesis, by disassembly of the PCNA/Polδ complex [9,78]. Remarkably, Esc2-mediated removal of Srs2 promotes Rad51 accumulation at the fork (Figure 3b) [77]. While this is more consistent with Srs2 disrupting Rad51 nucleofilaments than with it inhibiting DNA repair synthesis, it does not exclude the possibility that both activities operate at the fork.

## 5. From the ssDNA Gap at the Fork to the ssDNA Gap behind the Fork

Single-strand DNA fragments have been physically detected behind replication forks in organisms from yeast to humans, with a strong accumulation after cells are treated with genotoxic agents [20,21,32,34]. In principle, only those lesions occurring at the leading strand should block the advance of the polymerase, as continuous repriming during the synthesis of the lagging strand provides a simple way to bypass the lesion. This prediction has been validated by studies in bacteria and yeast both biochemically and in vivo [79,80,81,82]. However, ssDNA fragments accumulate at both strands [20,21,34], indicating that the process of lagging strand synthesis helps to bypass lesions but not to fill in the gaps. On the other hand, a mechanism is required for the replication fork to bypass the blocking lesion at the leading strand. This mechanism is unlikely to be associated with fork reversion, in which lesion bypass is thought to be associated with concomitant ssDNA gap filling (Figure 2). Mammalian cells possess a DNA polymerase with primase activity, PrimPol, that can prime new DNA synthesis downstream of a blocking lesion; indeed, primase-null PrimPol mutants are defective for progression through UV-damaged DNA, highlighting the physiological relevance of this mechanism for bypassing replication fork-blocking lesions in vivo [83,84]. In yeast, which lack a homolog of PrimPol, point mutations in the Polα/primase complex that have no effect on bulk DNA replication lead to an accumulation of forks with long ssDNA gaps and multiple internal gaps behind the fork in the presence of MMS, in addition to reversed forks and genomic rearrangements. This suggests that the replicative Polα/primase complex is required for repriming upon replication fork blockage (Figure 3c) [59]. The efficiency of the yeast replisome at priming DNA synthesis downstream of a blocking lesion is low in vitro but can be stimulated by RPA depletion, indicating that additional factors may modulate this process and they depend on the type and amount of DNA lesions [82].

## 6. Recombination behind the Fork

Although the post-replicative repair of ssDNA lesions is a conserved strategy to tolerate replicative lesions, the contribution of HR in different species is unclear. The absence of Rad51 and Rad52 causes an accumulation of ssDNA fragments behind the fork in yeast cells and *Xenopus* egg extracts treated with UV light or MMS [20,21,32], whereas Rad51 knockdown by RNA interference (RNAi) has no detectable effect on the amount of post-replicative ssDNA gaps in human cells treated with CPT, MMC, or HU [34]. This suggests a major role for TLS in gap-filling in human cells, in which up to 17 DNA polymerases were identified and characterized [13]. In line with this, the absence of the TLS polymerases Polη or Polζ in mammalian cells treated with UV light does not affect replication fork progression but causes an accumulation of post-replicative ssDNA gaps [85,86]. However, many factors and activities involved in post-replicative gap-filling by HR in yeast are conserved in mammalian cells (see below). One possibility is that the levels of Rad51 upon RNAi, although insufficient to promote fork reversal, are high enough to promote gap-filling [34]. Alternatively, the requirement for HR might be related to the type of DNA lesion. MMC-induced interstrand crosslinks, like the CPT-induced Top1cc, are hard-to-bypass lesions, and HU causes fork stalling by depleting the pool of available deoxynucleotides (dNTPs) [87]. In yeast, UV and MMS, but not HU, lead to ssDNA gaps behind the advancing forks [32], which are repaired not only by HR but also by TLS [21].

Therefore, most of our knowledge about the mechanisms by which post-replicative ssDNA gaps are repaired by HR comes from yeast. HR proteins fill in the ssDNA gaps using the information of the intact sister chromatid; this process leads to the formation of Rad51-dependent X-shaped sister chromatid junction (SCJ) structures that can be detected by 2D-gel electrophoresis in MMS-treated cells lacking the STR dissolvase [88]. Different genetic and molecular approaches, including analyses of SCJ formation, have elucidated many of the factors that cooperate with Rad51 in gap-filling by HR, providing a comprehensive view of the process. The first step is the enlargement of ssDNA gaps by the activities of the exonuclease Exo1, and to a lesser extent Mre11, and the helicase Pif1, through a process that is regulated by physical interactions with the 9-1-1 and PCNA clamps at the 5′- and 3′-junctions of the ssDNA gap, respectively (Figure 3d) [32,89]. A major role for Mre11 in expanding the post-replicative ssDNA gaps has been reported in *Xenopus* egg extracts [20]. ssDNA gap processing is required for (i) checkpoint activation by Rad9 and Rad53, which in turn downregulates Exo1 and Pif1 activity to prevent excessive and deleterious DNA degradation [32]; and (ii) further sister chromatid invasion, as inferred from the requirement of Exo1 and 9-1-1 for the accumulation of SCJs [90,91]. This strategy has also revealed the core recombination factors involved in the strand invasion step that leads to the formation of a D-loop intermediate: the complex RPA, the mediator Rad52, the multifaceted Rad54 helicase, and the helper complex Rad55/Rad57 (Figure 3e) [90,92,93]. A major difference between HR gap-filling and DSB recombinational repair is the dispensability of Rad59 and the assistance of an additional helper factor (Shu complex) in the former [90,94]. This requirement for the Shu complex is shared with the helicase Mph1 [95]. After strand invasion, the D-loop is extended by DNA synthesis using the intact sister chromatid, through a process that requires polymerase δ, whereby the polymerase ε and the TLS polymerases are dispensable (Figure 3e) [90]. Electron microscopy studies of the X-shaped structures that accumulate in response to MMS have recently helped to elucidate the different intermediates leading to SCJs. These analyses suggest that DNA invasion is mediated by annealing the ssDNA gap to the parental strand in the sister chromatid, rather than by the blocked 3′-end; annealing exposes the newly synthesized strand in the sister chromatid, which then acts as a template for DNA synthesis. This intermediate might be disassembled by the helicase activity of Sgs1, promoting synthesis-dependent strand annealing (SDSA) (Figure 3f,g), or alternatively, being captured by the 5′-end of the gap to form a double HJ-like structure with the biochemical features of a hemicatenane (Figure 3h) [56]. This model predicts topological constraints that would explain the requirement for sister chromatid cohesion and the DNA bending activity of the Hmo1 protein in SCJ formation [59,96]. Finally, SCJs are primarily dissolved by the STR complex through a mechanism that requires STR activation by sumoylation, which is performed by the Smc5/Smc6 complex together with the Ubc9 (E2)/Mms21 (E3) sumoylation enzymes (Figure 3h) [97,98,99,100,101]. Notably, Esc2 acts in concert with the STR complex, likely through its physical interaction with Ubc9 and SUMO [98].

However, the scenario is much more complex, as genetic analyses suggest the existence of two differentially regulated HR pathways dealing with ssDNA gaps. The process of SCJ formation described so far is termed template switching [14,15], and is dependent on PCNA polyubiquitylation (UbPCNA) at lysine 164 [74,93]. This modification occurs on chromatin in response to DNA damage by two E2 conjugase/E3 ligase ubiquitylation complexes: the Rad6 (E2)/Rad18 (E3) complex first monoubiquitylates lysine 164, and then the Mms2/Ubc13 (E2)/Rad5 (E3) complex extends this ubiquitylation with a K63-linked polyubiquitin chain (Figure 3b) [74,102,103]. Importantly, this UbPCNA-dependent HR pathway operates mostly in the S phase [104,105,106].

The sumoylation complex of E2 conjugase (Ubc9)/E3 ligase (Siz1) sumoylates chromatin-bound PCNA at lysine 164 during S phase not only under unperturbed conditions but also in response to replicative stress [74,107]. This is possible because PCNA is a homotrimer that can be sumoylated and ubiquitylated simultaneously [108]. PCNA sumoylation recruits the antirecombinogenic helicase Srs2, which prevents unscheduled HR. These recombination events can be detected in cells defective in both PCNA ubiquitylation and sumoylation/Srs2, where the HR proteins Rad51, Rad52, Rad54, and Rad55/Rad57, but not Rad59, provide resistance to MMS or UV light [75,76,109]; importantly, they also lead to the formation of SCJs that migrate in 2D-gels and are similar to those associated with the UbPCNA/HR pathway [93]. As PCNA sumoylation is restricted to the S phase [74], this UbPCNA-independent HR pathway (also termed salvage pathway) has been suggested to operate in G2/M. Moreover, the fact that Rad6/Rad18 binds to and activates preferentially sumoylated PCNA favors the UbPCNA/HR pathway during the S phase [108]. Therefore, the hub sumoylated PCNA/Srs2 controls HR both at and behind the fork: it inhibits HR at the fork during unperturbed DNA replication, and this inhibition is released from the fork and then transferred to the ssDNA gaps behind the fork in response to DNA damage.

It must be stressed that PCNA monoubiquitylation governs TLS working as an interacting platform for TLS polymerases [13]. Nevertheless, physical interactions of Rad5, or its human orthologs HLTF and SHPRH, with Rad18 and PCNA promote the extension of the polyubiquitin chain during S phase, postponing TLS to G2/M [74,110], at which point the levels of Rad5 decline and the TLS polymerase Rev1 reaches its maximal expression [106,111]. Importantly, most of the factors and activities leading to PCNA ubiquitylation and sumoylation, as well as the regulation of these processes, are conserved in mammalian cells [112], further supporting the idea that HR plays a role in the post-replicative filling-in of ssDNA gaps.

## 7. Controlling Recombination behind the Fork

The existence of a recombinational ssDNA gap-filling pathway that is inhibited during S phase raises two important questions: (i) How is this pathway controlled without affecting the UbPCNA/HR pathway? (ii) Why is this pathway inhibited during S phase? We do not have answers yet, although some data suggest that both questions are connected. As UbPCNA-dependent HR is active during the S phase, Srs2 may inhibit a step subsequent to strand invasion. In line with this idea, *srs2* mutants deficient for their helicase activity (and therefore deficient in displacing Rad51 from the nucleofilament) inhibit HR to the same degree as the wild-type Srs2 protein; however, a mutation in Srs2 that impairs its interaction with sumoylated PCNA, which is a prerequisite for Srs2-mediated disassembly of the PCNA/Polδ complex, no longer inhibits HR as inferred from its capacity to rescue the UV sensitivity of a *rad18* null mutant [78]. This *srs2* mutant cannot block the synthesis-dependent extension of D-loops in vitro, and accordingly, it displays longer conversion tracts and higher crossover frequencies than the wild-type [78]. Although crossovers between sister chromatids have no genetic consequences, they can lead to deletions, inversions, translocations, and loss of heterozigosity when occurring between ectopic or allelic homologous sequences [113]. Thus, cells preferentially dissolve the HJ-like structures to non-crossover products with the STR complex, restricting the activity of resolvases to late G2/M as a last option, as these enzymes also give rise to crossovers [114,115,116].

The UbPCNA/HR (template switching) and the UbPCNA-independent/HR (salvage pathway) are considered to be two distinct error-free DDT mechanisms [75,76,91,93]. This explains why a double mutant defective in PCNA ubiquitylation and HR is more sensitive to MMS or UV light than the single mutants [75]. This idea is also supported by the fact that UbPCNA/HR operates mostly in the S phase, and UbPCNA-independent/HR, in the G2/M phases. Indeed, the prominence of the dissolvase STR in the S phase, and the resolvases Mus81/Mms4 and Yen1 in G2/M, suggest that they process different SCJs at each pathway, which is also consistent with the specific role of Hmo1 in SCJ formation by the UbPCNA/HR pathway [96]. In line with this, SCJs with the properties of both hemicatenanes and HJ structures are detected in the presence of MMS (Figure 3h,i) [88,117]. HJs, but not hemicatenanes, can be processed by resolvases [12]; however, the expression of heterologous resolvases cannot prevent early accumulation of SCJs [117]. A possible explanation for these results is that the hemicatenanes formed via UbPCNA/HR that remain undissolved in G2/M or in PCNA sumoylation/*srs2* mutants are converted into HJs and become a substrate for resolvases [15].

These results are not incompatible with the possibility that, rather than an independent mechanism, the salvage pathway represents the loss of a regulatory mechanism aimed to prevent excessive DNA synthesis, high crossover frequencies, and genetic instability during the post-replicative repair of ssDNA gaps by the UbPCNA/HR pathway [78]. The helicase activity of Sgs1 would also contribute to disassembling the D-loop intermediate, thus favoring SDSA and non-crossover products (Figure 3f) [56]. This control would be removed at G2/M to accelerate the repair process before chromosome segregation. In this framework, Rad5-mediated polyubiquitylation would be required to counteract the inhibitory activity of Srs2 to ensure sufficient DNA synthesis to circumvent the blocking lesion, and to fine-tune the regulation of the D-loop extension step.

PCNA polyubiquitylation drives the DNA-dependent ATPase Mgs1 to HU-stalled replication forks [118], but it is unclear if this targeting also occurs at the ssDNA lesions left behind the fork or if it reflects the role of Rad5 in replication fork progression in the presence of replicative DNA damage [62,106]. Moreover, Mgs1 binding to polyubiquitylated PCNA interferes with the binding of Polδ to PCNA [118], suggesting an anti-recombinogenic role. Indeed, the absence of Mgs1 causes hyper-recombination [119]. In any case, the mild sensitivity to MMS of *mgs1* null mutants rules out Mgs1 as the essential target of PCNA polyubiquitylation in SCJ formation [119]. In human cells, PCNA polyubiquitylation interacts with the helicase ZRANB3, but again, it seems that this interaction occurs at stalled forks [120]. In any case, these results suggest that PCNA polyubiquitylation may serve as an interacting platform to factors that either actively promote the UbPCNA/HR pathway or counteract the Srs2 activity (Figure 3f).

It is important to recall at this point that Rad5 and HLTF have DNA strand exchange activities that, in contrast to Rad51, do not require ATP binding and/or hydrolysis [9]. Thus, although its helicase activity is dispensable for SCJ formation [121], Rad5 might help Rad51 to invade the intact sister chromatid (Figure 3e). Indeed, unlike Rad51, HLTF can use gapped DNA, such as that formed behind the fork, for D-loop formation in vitro [9].

Although less explored, evidence has begun to suggest that chromatin dynamics have a decisive role in regulating HR during post-replicative repair. This control operates at different levels: the chromatin remodeller Fun30 seems to act together with Exo1 in post-replicative ssDNA resection [122] as shown for DSB repair [123,124,125,126], whereas the chromatin assembly factor CAF1 counteracts the activity of Rqh1 (the Sgs1 ortholog in *Schizosaccharomyces pombe*) to stabilize the D-loop intermediate through physical contacts with PCNA [127]. Moreover, histone modifications ensure the completion and accuracy of the recombination events. Specifically, the ubiquitin ligase Bre1 accumulates at MMS-induced ssDNA lesions and contributes to SCJ formation by HR through H2B ubiquitylation at lysine 123 [128]. In addition, acetylation of histone H4 at lysine 16 by NuA4, through a process that additionally requires the chromatin remodeller RSC, promotes high fidelity and Rad5-dependent HR at CAG/CTG trinucleotide repeats that are prone to form hard-to-bypass hairpin structures [129].

## 8. Are the Recombination Processes at and behind the Fork Mechanistically Connected?

Yeast cells genetically modified to express Rad52 only in G2/M are defective not only in replication fork advance through damaged DNA, but also in the post-replicative filling-in of ssDNA lesions by HR. This defect is a consequence of the inability of these cells to load Rad51 at ssDNA lesions, indicating that recombinase binding to the ssDNA gaps left behind the fork is somehow coupled to DNA replication [23]. This is an important difference to the recombinational repair of DSBs, which can be completed in cells expressing Rad52 only in G2/M, as the binding of HR proteins to DSBs is replication-independent [23,25,26]. As Rad52 and Rad51 travel with the fork under unperturbed conditions, we proposed that repriming of DNA synthesis downstream of the DNA blocking lesion would leave the recombination proteins loaded at the ssDNA gap left behind the fork (Figure 3c) [23]. In agreement with this expectation, Polα/primase mutants deficient in repriming DNA synthesis after the blocking lesion accumulate post-replicative ssDNA gaps whose repair is associated with high levels of TLS-mediated mutagenesis and genomic rearrangements, but which are defective in SCJ formation [59]. Therefore, proper DNA repriming by the Polα/primase complex is required for efficient post-replicative recombinational processing of ssDNA gaps. Additional evidence supports the idea that the recombinational post-replicative filling-in of ssDNA gaps is coupled to replication fork dynamics. Defective SCJ formation in mutants that cannot ubiquitylate H2B at K123 is associated with a loss of replication intermediates, which is likely due to the role of this modification in chromatin assembly of newly synthesized histones and replisome stability [130]. Likewise, the translocase Irc5 facilitates replication fork advance through alkylated DNA, SCJ formation, and recombinational gap filling by assisting in the enrichment of cohesion complexes at stalled forks [131].

If recombination initiation through the loading of HR proteins were to be coupled to replication fork stalling and DNA synthesis repriming, not only the proteins travelling with the fork but also those recruited in response to DNA damage would get directly loaded at the ssDNA gaps left behind the fork as DNA synthesis resumes. Further recombination steps, including resection, D-loop formation, extension, and disassembly, as well as SCJ formation and dissolution/resolution, may occur post-replicatively to facilitate the completion of genome replication. This model would mechanistically link the activity at and behind the fork of many of the factors involved in DDT, although it does not exclude the possibility that some of these can also access the recombinogenic intermediates post-replicatively, as has been shown for Rad18, Rad5, Sgs1, or Smc5/Smc6 [99,104,105].

## 9. Concluding Remarks

Genome duplication opens a window in which DNA is particularly susceptible to undergoing mutations and rearrangements. The replication fork accumulates free DNA ends and ssDNA fragments, while continuous nucleosome disassembly leaves DNA more accessible to nucleases and DNA processing activities [132]. This scenario is particularly aggravated by endogenous and exogenous agents that hamper replication fork advance. This explains the relevance that HR has during DNA replication to prevent genetic instability and the association between defective recombination factors with cancer and many genetic diseases [133]. Intensive research as well as novel and powerful tools have strongly improved our knowledge about the mechanisms by which HR deals with replicative stress, which may help to develop new therapeutic strategies. Still, many open questions need to be addressed, some of which were already outlined in this review. What is the contribution of HR to ssDNA gap-filling in mammalian cells, especially in response to DNA damaging agents like MMS or UV light? Are the recombination processes at and behind the fork mechanistically coupled, and if so, how? Is the salvage pathway a genetically distinct mechanism? Do HR proteins operate during DDT exclusively through their recombinogenic activities? Are the processes of HR and TLS interconnected, or they are independently regulated? These questions require the development of new approaches to dynamically follow the activity of the recombination proteins along the DNA and through the cell cycle, as well as new separation-of-function mutations that help to mechanistically dissect these processes. However, addressing these questions is sure to bring novel and exciting advances to the field.

## Figures and Tables

**Figure 1 genes-09-00603-f001:**
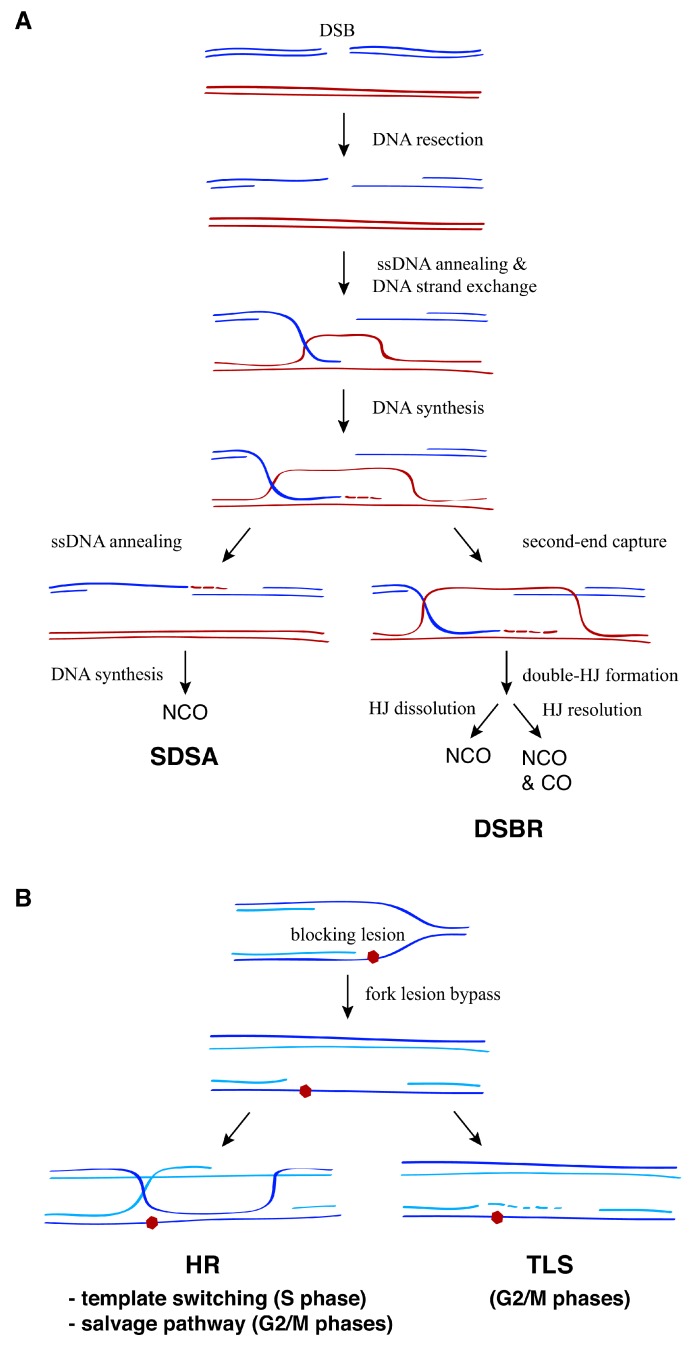
General strategies for the recombinational repair of double-strand breaks (DSBs) and replication associated single-stranded DNA (ssDNA) gaps. (**A**) In response to DSBs, the 5′-ends of the break are resected, leading to 3′-ended ssDNA molecules that search for and then invade a homologous DNA sequence through Rad51-dependent ssDNA annealing and DNA strand exchange reactions. This generates a D-loop intermediate that can be processed by two major pathways: synthesis-dependent strand annealing (SDSA) or double-strand break repair (DSBR). Whereas SDSA leads to non-crossovers (NCO), the output of DSBR (non-crossover versus crossover) depends on whether the double-Holliday junction (HJ) is dissolved by the STR complex or resolved by DNA structure-specific nucleases. (**B**) DNA damage tolerance (DDT) mechanisms promote replication fork advancement through DNA lesions that hamper DNA synthesis, postponing the repair of the blocking lesion for later. This process can lead to the formation of ssDNA fragments behind the fork that are repaired post-replicatively by either translesion synthesis (TLS) or homologous recombination (HR). Two HR pathways have been proposed to operate depending on the cell cycle phase: template switching during the S phase and the salvage pathway during the G2/M phases.

**Figure 2 genes-09-00603-f002:**
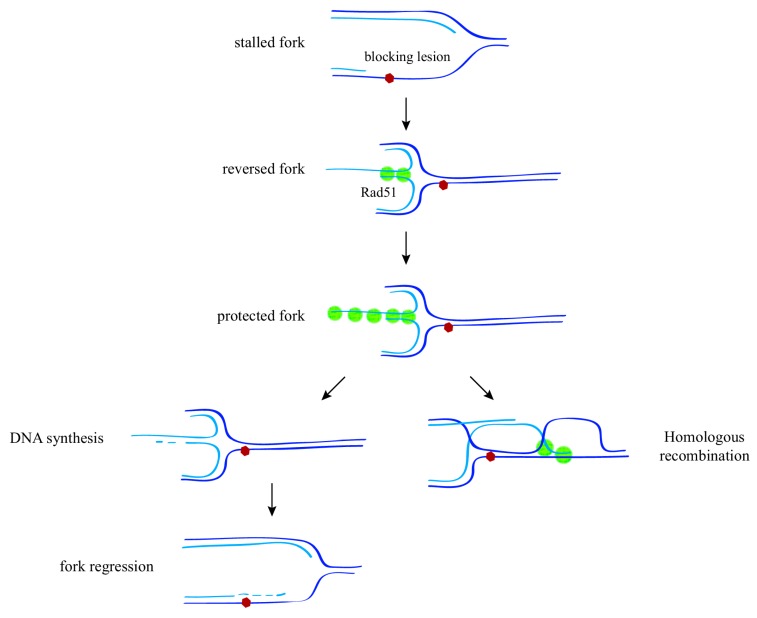
Mechanisms of replication fork protection and restart upon replication stress. In response to replication stress (e.g., a blocking lesion that uncouples DNA unwinding from DNA synthesis), displacement and further reannealing of the nascent strands leads to the formation of a reversed fork through a process that requires the recombination protein Rad51. Stabilization of a Rad51 nucleofilament at this structure is required to protect the fork against nuclease degradation. Reversed forks might facilitate lesion bypass by either DNA synthesis and fork regression, or strand invasion ahead of the fork, thus reducing the need for post-replicative ssDNA repair.

**Figure 3 genes-09-00603-f003:**
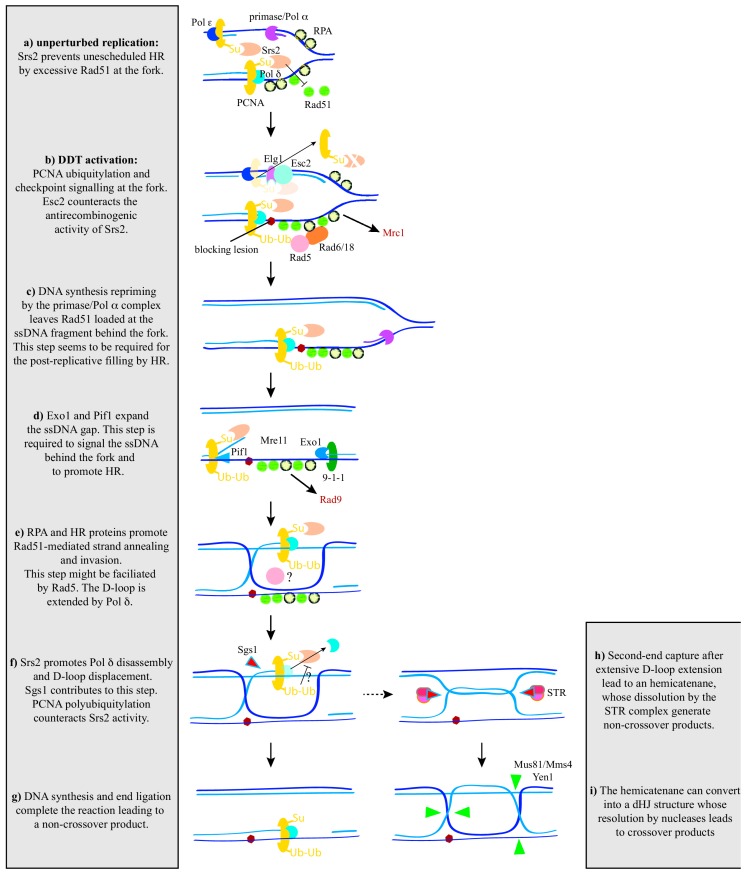
A unifying model for recombination activities at and behind the fork in yeast. See text for details. PCNA: proliferating cell nuclear antigen; RPA: replication protein A.

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
