# Peer review of "Homologous Recombination: To Fork and Beyond"

_genes, 2018, doi:10.3390/genes9120603_

Round 1
Reviewer 1 Report
This review by Felix Prado entitled "Homologous recombination: To fork and beyond" addresses essential role of this mechanism required to maintain genome stability associated with replication stress. The author provides comprehensive overview of this process and I really enjoyed reading it. I recommend it for publishing but have stressed some minor points that the review would benefit after addressing.
Here I suggest few points that in my view should be discussed:
· While role of EXO1 has been discussed, involvement of other nucleases, in particular MRE11 should be discussed.
· The review would also benefit from commenting on translocases required for fork reversal and fork restart. Regarding antirecombinases, RECQ5 and/or FBH1 seems to be more direct functional homologue of yeast Srs2 and should be mentioned accordingly.
· Fanconi anomia (FA) group of proteins is also described to play important role in dealing with replication stress and some factors modulated and co-localize with HR proteins and some even play in HR direct role.· It is clear that RAD51 is an essential component for majority of the steps from fork protection, to fork reversion, as well as fork restart. This might reflect the dynamic /structural features of the filament as well as identification of numerous accessory proteins.
· I would also recommend to further stress the checkpoint response to replication stress.
· In general more figures would also help readers outside the field to understand described mechanism. Clearer description of “salvage”, template switch, BIR and gap repair pathways would be also appreciated in this context.
· In the introduction the author describe RAD51 is a sole protein capable of strand change however RAD52 and HLTF were also described to promote strand invasion and homology search.
Author Response
1. Comments and Suggestions for Authors
This review by Felix Prado entitled "Homologous recombination: To fork and beyond" addresses essential role of this mechanism required to maintain genome stability associated with replication stress. The author provides comprehensive overview of this process and I really enjoyed reading it. I recommend it for publishing but have stressed some minor points that the review would benefit after addressing.
Here I suggest few points that in my view should be discussed:
· While role of EXO1 has been discussed, involvement of other nucleases, in particular MRE11 should be discussed.
Done as suggested; the role of Mre11 at and behind the fork is commented, as well as the Dna2 nuclease (lines 131-134, 260-265 and 190-193) (for comparison their role in DNA resection during DSB repair is outlined when introducing this point in the first paragraph; lanes 33-34).
· The review would also benefit from commenting on translocases required for fork reversal and fork restart.
Those translocases have been mentioned in the new version (lines 126-128). I have not extended the details about them because it goes beyond the scope of this review, and referred to two recent and detailed reviews for the process of fork reversal and restart.
Regarding antirecombinases, RECQ5 and/or FBH1 seems to be more direct functional homologue of yeast Srs2 and should be mentioned accordingly.
The antirecombinogenic role of these enzymes have been included and discussed together with the one by Srs2 and Sgs1 to better explain the mechanisms of HR control at the fork (lines 187-193)
· Fanconi anomia (FA) group of proteins is also described to play important role in dealing with replication stress and some factors modulated and co-localize with HR proteins and some even play in HR direct role.
The idea that HR plays replicative roles in the FA pathway has been included during the discussion of the role of HR in response to different types of replicative lesions (lines 134-138). This is a specific pathway that shares common elements with the mechanisms described here, but that requires a separate analysis due to its singularity. A recent review discussing the interconnection between HR and FA is cited.
· It is clear that RAD51 is an essential component for majority of the steps from fork protection, to fork reversion, as well as fork restart. This might reflect the dynamic /structural features of the filament as well as identification of numerous accessory proteins.
I hope this idea gets clearer in the revised manuscript; however, as mentioned before, the specific mechanisms by which translocases operate at forks is beyond the scope of this review, and recent and excellent reviews about this topic are cited.
· I would also recommend to further stress the checkpoint response to replication stress.
Done as suggested (see lines 84-90)
· In general more figures would also help readers outside the field to understand described mechanism. Clearer description of “salvage”, template switch, BIR and gap repair pathways would be also appreciated in this context.
I have included a new Figure (1A) to briefly describe the essential steps in the recombinational repair of DSBs, which may help people that is not familiarized with HR to follow its role in DDT. BIR is not described as it is not related to DDT and it might be confusing, although it is indicated in the legend of Figure 1A that DSBs can be also processed by BIR and SSA. In new Figure 1B the major DDT strategies are mentioned to be compared with the recombinational repair of DSBs. The details of the template switching and salvage pathways are explained in the text in section 6 (recombination behind the fork).
· In the introduction the author describe RAD51 is a sole protein capable of strand change however RAD52 and HLTF were also described to promote strand invasion and homology search.
This has been corrected (lines 34-37) and the biological relevance of this role discussed later (lines 376-380)

Reviewer 2 Report
“Homologous Recombination: to fork and beyond” by Felix Prado
In this review Felix Prado elaborates a model for how factors of the homologous recombination pathway act to on the one hand ensure protection of stalled replication forks/lesion bypass during S phase and promote HR on single-stranded gaps behind replication forks in a post-replicative manner. A particular focus in the review is the budding yeast system, where the author utilizes the full breadth of published data to bring it all into a consistent model.
I was very impressed by this review and the authors attempt to present a unifying picture derived from many different studies. To me this is entirely convincing, I think it is an excellent piece that hopefully will influence the thinking of many researchers. As such, I have only minor things to add.
(1) Ubiquitin-modification should be referred to either as ubiquitination or ubiquitylation, not ubiquitilation.
(2) At least in my pdf all greek letters are not displayed correctly, which makes it difficult to discriminated the different polymerases.
(3) line 298: perhaps change to “mutants deficient for their helicase activity and therefore deficient in displacing Rad51”
(4) Maybe additional citations would be helpful for regulation of HR by the Fun30 nucleosome remodeller (e.g. Costelloe et al; Chen et al (2 papers); Eapen et al; Bantele et al.)
(5) Reference 103 lacks a journal title.
Author Response
2. Comments and Suggestions for Authors
“Homologous Recombination: to fork and beyond” by Felix Prado
In this review Felix Prado elaborates a model for how factors of the homologous recombination pathway act to on the one hand ensure protection of stalled replication forks/lesion bypass during S phase and promote HR on single-stranded gaps behind replication forks in a post-replicative manner. A particular focus in the review is the budding yeast system, where the author utilizes the full breadth of published data to bring it all into a consistent model.
I was very impressed by this review and the authors attempt to present a unifying picture derived from many different studies. To me this is entirely convincing, I think it is an excellent piece that hopefully will influence the thinking of many researchers. As such, I have only minor things to add.
(1) Ubiquitin-modification should be referred to either as ubiquitination or ubiquitylation, not ubiquitilation.
Corrected as requested
(2) At least in my pdf all greek letters are not displayed correctly, which makes it difficult to discriminated the different polymerases.
Corrected.
(3) line 298: perhaps change to “mutants deficient for their helicase activity and therefore deficient in displacing Rad51”
Changed as suggested
(4) Maybe additional citations would be helpful for regulation of HR by the Fun30 nucleosome remodeller (e.g. Costelloe et al; Chen et al (2 papers); Eapen et al; Bantele et al.)
Included as suggested
(5) Reference 103 lacks a journal title.
Corrected

Reviewer 3 Report
This MS is a thorough review of the current literature regarding repair/processing of ssDNA gaps at and behind the replication fork by homologous recombination. The review is well focused, which allows the author to provide a thorough, organized review of the current state of this field.
Overall, this is a good and clear review; minor clarifications are suggested below.
Whole text: There are several English errors, some of which change the meaning of sentences. Specific cases are indicated below, but revision of the whole text is suggested to ensure that sentences are clear.
Whole text: All of the Greek symbols (e.g., in the names of polymerases) appear as the same symbol in my copy; this may be an issue of compatibility between word processors.
Specific points:
Line 47: The author defines the abbreviation TS as “template switching.” This abbreviation is only used once, on line 263. I would suggest the abbreviation not be used for the sake of increased clarity for the reader.
Line 67: The sentence immediately prior does not really indicate that “…HR deals with ssDNA lesions both at and behind the fork”.
Line 68: “Indeed, analyses of DNA damage-induced Rad51 containing foci in yeast and mammalian cells showed that they represent non-replicative, DNA repair centers [13,14] that at least in the case of yeast are restricted to G2/M through a Mrc1-dependent mechanism that prevent their assembly during S phase [14,15].”
This sentence is somewhat misleading. In the cited article (EMBO J. 2013, 32, 1307–1321, figure 3C), Rad51 foci appear to form in S phase, although Rad52 foci are more clearly restricted to G2/M. This should be clarified. Perhaps work by P. Pasero’s (PMID: 19322196) and R. Rothstein’s (PMID: 19262568) groups should also be cited here as well.
Line 91: “However, human Rad51 slows down DNA synthesis [21,25], whereas yeast Rad51 accelerates it [14,26,27], suggesting different but not yet known modes of action on blocked forks.”
This sentence is peculiar since in the next paragraphs the author explains several actions of Rad51 that could explain at least in part these observations. I would suggest to remove "but not yet known" from the sentence, or replaced by "incompletely understood"?
Line 124: “Since the mediators BRCA2 and Rad52 facilitate Rad51 binding at ssDNA by competing with RPA…”
This sentence is unclear: which of these three proteins compete with RPA? Presumably Rad51?
Line 132: This line refers to “the absence,” but doesn’t state of what (presumably “of fork reversal”). If so, add the word “of” after “the absence”.
Line 153: Thus far, we lack of information about this function, but it is tempting to speculate with the possibility that Rad51 to be required to couple DNA unwinding and synthesis, being this function more critical in the presence of DNA damage.
This sentence is unclear and should be modified.
Line 279: “Moreover, the fact that Rad6/Rad18 binds to and activates preferentially sumoylated PCNA favors the UbPCNA/HR pathway during S phase…”
This sentence is counterintuitive since the same residue is ub or sumoylated. This should be clarified.
Line 351: The author refers to “histone acetylation at lysine 16,” but not which histone; this should be specified.
Line 371: The word “Thus” should be removed, as it hinders the understanding of the second half of this paragraph.
Author Response
3. Comments and Suggestions for Authors
This MS is a thorough review of the current literature regarding repair/processing of ssDNA gaps at and behind the replication fork by homologous recombination. The review is well focused, which allows the author to provide a thorough, organized review of the current state of this field.
Overall, this is a good and clear review; minor clarifications are suggested below.
Whole text: There are several English errors, some of which change the meaning of sentences. Specific cases are indicated below, but revision of the whole text is suggested to ensure that sentences are clear.
Revised by a professional English editing service
Whole text: All of the Greek symbols (e.g., in the names of polymerases) appear as the same symbol in my copy; this may be an issue of compatibility between word processors
Corrected
Specific points:
Line 47: The author defines the abbreviation TS as “template switching.” This abbreviation is only used once, on line 263. I would suggest the abbreviation not be used for the sake of increased clarity for the reader.
Template switching is now used without abbreviation in the few occasions is mentioned
Line 67: The sentence immediately prior does not really indicate that “…HR deals with ssDNA lesions both at and behind the fork”.
It is true that those results refer only to events behind the fork. For clarity it has been changed to: “HR also operates at ssDNA lesions behind the fork”
Line 68: “Indeed, analyses of DNA damage-induced Rad51 containing foci in yeast and mammalian cells showed that they represent non-replicative, DNA repair centers [13,14] that at least in the case of yeast are restricted to G2/M through a Mrc1-dependent mechanism that prevent their assembly during S phase [14,15].”
This sentence is somewhat misleading. In the cited article (EMBO J. 2013, 32, 1307–1321, figure 3C), Rad51 foci appear to form in S phase, although Rad52 foci are more clearly restricted to G2/M. This should be clarified. Perhaps work by P. Pasero’s (PMID: 19322196) and R. Rothstein’s (PMID: 19262568) groups should also be cited here as well.
This has been clarified; now it reads: “Indeed, analyses of replicative DNA damage-induced Rad51 containing foci in yeast and mammalian cells showed the existence of both replicative and non-replicative, DNA repair centers {Petermann:2010jw, GonzalezPrieto:2013go}, with the latter restricted to G2/M in yeast through a Mrc1-dependent mechanism that prevent their assembly during S phase {Meister:2005ea, Alabert:2009gh, Barlow:2009jl, GonzalezPrieto:2013go}.” The indicated references have been included as suggested.
Line 91: “However, human Rad51 slows down DNA synthesis [21,25], whereas yeast Rad51 accelerates it [14,26,27], suggesting different but not yet known modes of action on blocked forks.”
This sentence is peculiar since in the next paragraphs the author explains several actions of Rad51 that could explain at least in part these observations. I would suggest to remove "but not yet known" from the sentence, or replaced by "incompletely understood"?
Removed as suggested
Line 124: “Since the mediators BRCA2 and Rad52 facilitate Rad51 binding at ssDNA by competing with RPA…”
This sentence is unclear: which of these three proteins compete with RPA? Presumably Rad51?
Clarified; now it reads: “Since the mediators BRCA2 and Rad52 help Rad51 to compete with RPA at ssDNA …”
Line 132: This line refers to “the absence,” but doesn’t state of what (presumably “of fork reversal”). If so, add the word “of” after “the absence”.
Corrected; now it reads: “both the lack of and an excess of fork reversion”
Line 153: Thus far, we lack of information about this function, but it is tempting to speculate with the possibility that Rad51 to be required to couple DNA unwinding and synthesis, being this function more critical in the presence of DNA damage.
This sentence is unclear and should be modified.
The sentence has been changed to explain how Rad51 might prevent the accumulation of ssDNA. Now it reads: “Although we still lack information about this function, it is tempting to speculate about the possibility that Rad51 prevents ssDNA from accumulating at the fork by coupling DNA unwinding and DNA synthesis”.
Line 279: “Moreover, the fact that Rad6/Rad18 binds to and activates preferentially sumoylated PCNA favors the UbPCNA/HR pathway during S phase…”
This sentence is counterintuitive since the same residue is ub or sumoylated. This should be clarified.
This has been explained by stressing at the beginning of the paragraph that PCNA sumoylation “This is possible because PCNA is a homotrimer that can be sumoylated and ubiquitylated simultaneously {Parker:2012gw}” (lines 301-302)
Line 351: The author refers to “histone acetylation at lysine 16,” but not which histone; this should be specified.
Corrected; now it reads “acetylation of histone H4 at lysine 16 by NuA4”
Line 371: The word “Thus” should be removed, as it hinders the understanding of the second half of this paragraph.
Removed as suggested
